# Genotypic diversity and plasticity of root system architecture to nitrogen availability in oilseed rape

Christophe Lecarpentier[1,2]*, Loïc Pagès[1], Céline Richard-Molard[2]

1 INRAE, UR1115, Plantes et Systèmes de culture Horticoles, Clermont-Ferrand, France, 2 UMR ECOSYS, INRAE, AgroParisTech, Université Paris-Saclay, Thiverval-Grignon, France

* christophe.lecarpentier@inrae.fr

**Data Availability Statement:** All data files are available from the INRAE database at the following URL: https://data.inrae.fr/dataset.xhtml?persistentId=doi:10.15454/9WVGUH DOI of the

## Abstract

In the emerging new agricultural context, a drastic reduction in fertilizer usage is required. A promising way to maintain high crop yields while reducing fertilizer inputs is to breed new varieties with optimized root system architecture (RSA), designed to reach soil resources more efficiently. This relies on identifying key traits that underlie genotypic variability and plasticity of RSA in response to nutrient availability. The aim of our study was to characterize the RSA plasticity in response to nitrogen limitation of a set of contrasted oilseed rape genotypes, by using the ArchiSimple model parameters as screening traits. Eight accessions of *Brassica napus* were grown in long tubes in the greenhouse, under two contrasting levels of nitrogen availability. After plant excavation, roots were scanned at high resolution. Six RSA traits relative to root diameter, elongation rate and branching were measured, as well as nine growth and biomass allocation traits. The plasticity of each trait to nitrogen availability was estimated. Nitrogen-limited plants were characterized by a strong reduction in total biomass and leaf area. Even if the architecture traits were shown to be less plastic than allocation traits, significant nitrogen and genotype effects were highlighted on each RSA trait, except the root minimal diameter. Thus, the RSA of nitrogen-limited plants was primarily characterised by a reduced lateral root density, a smaller primary root diameter, associated with a stronger root dominance. Among the RSA traits measured, the inter-branch distance showed the highest plasticity with a level of 70%, in the same range as the most plastic allocation traits. This work suggests that lateral root density plays the key role in the adaptation of the root system to nitrogen availability and highlights inter-branch distance as a major target trait for breeding new varieties, better adapted to low input systems.

## 1. Introduction

Over recent decades, the agricultural paradigm has prioritized increasing yield year after year [1] over other ecosystem services, such as environmental concerns [2]. This has resulted in the massive use of agro-chemical fertilizers, among which nitrogen (N) fertilizers appear to be predominant with 120 Mt on a fertilizers global consumption of 200 Mt in 2018. Nitrogen is

dataset: Lecarpentier, Christophe, 2021, "Root system architecture plasticity to nitrogen availability for 8 genotypes of oilseed rape", https://doi.org/10.15454/9WVGUH, Portail Data INRAE, V1.

**Funding:** This work was funded by the French programme "Investments for the Future" (ANR-11-BT-BT-004 440 "RAPSODYN"). The funders had no role in study design, data collection and analysis, decision to publish, or preparation of the manuscript.

**Competing interests:** The authors have declared that no competing interests exist.

indeed one of the most required minerals in plant nutrition, as it is essential to crop growth and yield [3]. However, nitrogen fertilizers also have harmful effects on the environment [4], as a source of water pollution due to nitrate leaching [5] and air pollution due to N-derived greenhouse gas emissions [6]. Therefore, reducing the use of nitrogen fertilizers while maintaining high yields is emerging as one of the major challenges of the new agroecological paradigm [7, 8].

Root system plays a major role in N acquisition, both because of its absorption capacities and its soil exploration potential. Therefore, a better understanding of the adaptation of the root system architecture (RSA) to N availability appears to be a promising lever towards the optimization of N acquisition [9, 10]. However, the consideration of RSA into breeding programs is still emerging [11] and needs further investigation. As a first step, this requires identifying the key traits underlying genotype variability and plasticity of RSA to N availability.

RSA is defined as the result of the topology and spatial distribution of the root system [12]. It emerges from several root traits such as root number, root length and the angle between consecutive roots [13]. The combination of these traits results in a complex root system structure, which is difficult to phenotype simply, especially because the complexity increases with plant age. Local variations in hydric and mineral resources also increases RSA variability [14]. Therefore, most studies focus on a limited number of integrative RSA traits, such as root system biomass or total root length [9]. Some recent studies phenotype some analytical RSA traits, such as primary or lateral root length, number of lateral roots, root density, root area or specific root length, for which phenotyping capacities are most of the time limited to seedings [15, 16] or young plants [17–19] under controlled environmental conditions. However, as the root system has also been identified as a recursive structure, it should be possible to define relevant systemic traits that determine RSA independently of the position of the considered segment within the root system. Modeling approaches aim to simulate complex phenotypes throughout plant development, from a few time-constant analytical parameters. Therefore, a model-assisted phenotyping approach, in which the experimental study would be devoted to phenotyping the variation of the parameters of an RSA model in response to N could be an effective way to characterize RSA plasticity throughout the plant life while limiting the impact of uncontrolled environmental variation. This promising approach has however never been used yet to characterize RSA response to N availability.

The ArchiSimple model, proposed by Pagès et al. [20], provides an interesting framework to characterize RSA variability, as model parameters stand for analytical RSA traits, such as root diameter [21–23], branching density [10, 24] or elongation [21, 22]. The interspecific diversity of some of the ArchiSimple model parameters has formerly been investigated on more than 150 species including monocotyledonous and dicotyledonous [20, 25, 26], showing that root diameter varies primarily with species, while root hierarchy and branching vary primarily with environment. Although genotypic variability of RSA has been shown on integrative RSA traits [27], intraspecific diversity of ArchiSimple parameters has never been evaluated.

In addition, little work has investigated the effect of N availability on RSA [28, 29]. Some studies have focused on the effect of heterogeneous N distribution, provided by nitrogen-rich patches or strips [13, 30, 31], while others have analysed homogeneous N limitations [29, 32–34]. Results indicate that RSA modifications differ between homogeneous and heterogeneous N availabilities. Under heterogeneous N availability, modifications of individual root elongation, total root length, root number, root angles and lateral branching are observed when roots encounter a nitrogen-rich patch (see [31] or [35] for a review). In contrast, homogeneously N-limited environments do not induce changes in primary or lateral root elongation [13, 36]. This discrepancy suggests that the regulation of response processes differ between patchy

*versus* homogeneous N distribution. Furthermore, although previous works have highlighted the genetic variability of some RSA traits such as root diameters [25, 26], little is known about the relevant traits for determining RSA ideotypes adapted to limited N availability. Therefore, it appears necessary to characterize the plasticity in response to N availability in addition to genotypic diversity of the RSA traits, in order to identify patterns of RSA adaptation and select architectures with better foraging performances that are likely to be more performant under homogeneous low N environment [37].

In this paper, using a model-assisted phenotyping approach based on the ArchiSimple model, we aimed at identifying the relevant traits underlying RSA genotypic diversity and plasticity in response to homogeneous low N availability. Our study was carried out on a diversified panel of oilseed rape genotypes, this species being one of the crops with the highest N requirements per unit of yield produced [38] and knowing to be highly plastic in response to environmental constraints [18].

## 2. Materials and methods

### 2.1 Experimental design

Eight genotypes of *Brassica napus* were selected from a panel of 200 genotypes previously grown in the field under 20 combinations of location and year [39], to maximize (i) the phenotypic diversity of aerial architecture, (ii) the diversity of yield performance and (iii) sensitivity to the environment (*ie*. ability to produce different yields under various location x year conditions). The set of selected genotypes was also designed to represent the genetic diversity of oilseed rape in terms of registration year (1981–2003), thus accounting for genetic progress; oilseed rape type (winter or spring), which differ by bisannual or annual growth cycle; and finally seed quality according to their glucosinolate and erucic acid content (Table 1). Plants were grown in a greenhouse located in Avignon, southeastern France, on 32 tubes 10 cm diameter and 1 m long, filled with a 50/50 (v/v) mixture of thin pozzolan (< 5 mm) and thin vermiculite. Position of the tubes in the experimental setup was randomized. Eight seeds were sown in each tube on the 8th March 2018. Seedlings were thinned sixteen days after sowing, when plants had two unfolded leaves, so that only two plants remained in each tube on the 24th of March (phenological stage BBCH 12 [40]). Each tube was irrigated every other day with 200 mL of a modified Hoagland solution [41] supplemented with oligo-elements, iron and

**Table 1. Name and characteristics of the eight oilseed rape genotypes chosen for our experiment.**

| Genotype name | Oilseed rape type | Geographical origin | Registration year | Seed quality | Branching | Yield (t.ha$^{-1}$) | Yield variation |
|---|---|---|---|---|---|---|---|
| AMBER | Winter | Germany | 2003 | 0+ | 0/3 | 3.02 | 33% |
| AVISO | Winter | Denmark | 2000 | 00 | 3/3 | 3.37 | 18% |
| CRESOR | Spring | France | <1982 | 0+ | NA | NA | NA |
| EMIL | Winter | Denmark | 1981 | 0+ | 3/3 | 2.43 | 29% |
| GASPARD | Winter | France | 1985 | ++ | NA | 2.47 | 20% |
| MOHICAN | Winter | France | 1995 | 00 | 1/3 | 3.05 | 22% |
| MILENA | Winter | Germany | 2000 | 00 | 0/3 | 3.31 | 27% |
| TOSCA | Winter | Sweden | Early 2000's | 00 | NA | 2.55 | NA |

Seed quality is a qualitative indicator for erucic acid (C22:1) and glucosinolates (GSL) content, with ++: genotypes rich in C22:1 and GSL, 0+: genotypes poor in C22:1 and rich in GSL, and 00, genotypes poor in C22:1 and in GSL. The branching notation comes from previous experiments, ranging from 0/3 (very few aerial branches) to 3/3 (many aerial branches). Yield data indicates if the genotype is known for its high or low productivity, whereas yield variation data indicates whether or not the genotype had contrasting values of yield in various environments (ie. location x year), expressed by the coefficient of variation of yield among environments [39]. NA: not available. Yield data were extracted from the RAPSODYN project database, courtesy of Anne Laperche and Erwan Corlouer).

nitrate, that provided either 10 (N+) or 1 (N-) mEq of N (S1 Table). The N+ and N- solutions differed only in N concentration and were used to discriminate two contrasting levels of N availability among the tubes. Thus, at the end of the experiment, the two plants within each of the 16 N+ and 16 N- tubes had received an average of 0.56 g and 0.056 g of N, respectively.

## 2.2 Excavation procedure

The excavation procedure began at 800 growing degree-days after sowing (about 30 days) and lasted about 150 growing degree-days. The excavation order of the tubes was randomized. The substrate of each tube was excavated by a complete immersion into a water bath. The two plants of each tube were then collected in a sieve and the maximal length of the entangled root systems was measured immediately. Two kinds of samples were collected: (i) 30 cm samples of thick roots at the bottom of the tube (*i.e.* young parts of primary roots), (ii) samples of thin roots at the top of the tube (*i.e.* old roots). The samples collected were stored in water in a 4˚C cold chamber, pending further measurements.

The leaves of each plant were separated from the remaining roots and scanned using a flat-bed scanner (EPSON perfection V800 and V850) at a resolution of 600 dots per inch using the opaque mode. Individual plant leaf area was then estimated by analysing the scan with the ImageJ software (https://imagej.nih.gov/ij/) to compute the green area of all leaves for each plant. Leaves and remaining roots were then weighed, before and after drying in an oven at 70˚C for 72h to estimate wet and dry biomasses. Each plant compartment (roots and shoot) was then ground to a fine powder and subsequently analysed for carbon and nitrogen content according to the Dumas combustion method [42], using an automated CN analyser (FLASH EA 1112 Series).

## 2.3 Scanning and on-screen measurements of root samples

Root samples were carefully disentangled and spread in a transparent plastic tray containing a thin layer of water, avoiding root overlaps. The spread-out root samples were then scanned using a flatbed scanner (EPSON perfection V850) at a resolution of 2400 or 3200 dots per inch, using the transparent mode. The scans were then analysed manually on the computer screen with the ImageJ software to capture root traits [37]. On each scan of thick root samples, 50 measurements of the mother root diameter, lateral root diameter, and distance between two consecutive branches were carried out along the principal roots. Moreover, on each scan of thin root samples, the diameter of 30 of the thinnest roots was measured, as well as the diameter and length of 50 randomly chosen individual roots.

## 2.4 Trait definition

The previously described measurements led to the definition of 15 traits characterized at 800 growing degree-days: nine plant growth and allocation (PA) traits (TDB, RDB, LA, RS, RTD, NUpE, NUtE, CC, NC) and six root system architecture (RSA) traits (Dmin, Dmax, DlDm, IBD, VarD, ELT) (Table 2). The leaf area (LA) is the sum of the areas of all leaves of each individual plant. Plant biomass was described by two traits: total dry biomass (TDB) (the sum of dry biomasses of the below-ground part and the above-ground part of the plant) and root dry biomass (RDB). The Root:Shoot ratio (RS) is the ratio of the dry biomasses of the below-ground to the above-ground part of the plant. The root tissue density (RTD) is the ratio of the dry to the wet biomass of the below-ground part of the plant. This measurement represents a proxy of dry biomass over volume [43]. The total carbon content of the plant (CC) is the sum of the carbon content of each plant compartment (root or shoot) weighed and the ratio between the dry biomass of the appropriate compartment and the total biomass of the plant.

**Table 2. List of all the traits measured or calculated at the end of the experiment.**

| Trait | Type | Description | Unit |
|---|---|---|---|
| TDB | Allocation | Dry Biomass of the whole plant | g |
| RDB | Allocation | Dry Biomass of the root system | g |
| LA | Allocation | Leaf Area of all leaves of the plant | $cm^2$ |
| RS | Allocation | Root/Shoot dry biomass ratio | $g.g^{-1}$ |
| RTD | Allocation | Root Tissue Density: ratio between the dry and wet biomass of roots | $g.cm^{-3}$ |
| NUpE | Allocation | Nitrogen Uptake Efficiency | $g.g^{-1}$ |
| NUtE | Allocation | Nitrogen Utilization Efficiency | $g.g^{-1}$ |
| CC | Allocation | Carbon content | $g.100g^{-1}$ |
| NC | Allocation | Nitrogen content | $g.100g^{-1}$ |
| Dmin | RSA | Minimal diameter | mm |
| Dmax | RSA | Maximal diameter | mm |
| IBD | RSA | Inter-branch distance | mm |
| DlDm | RSA | Slope of diameter of lateral roots against diameter of their mother root | $mm.mm^{-1}$ |
| VarD | RSA | Variation coefficient of lateral roots diameter | – |
| ELT | RSA | Ratio of the elongation rate to the maximal diameter of the primary root. | $mm.mm^{-1}.°Cd^{-1}$ |

In the same way, the total nitrogen content of the plant (NC) is the sum of root nitrogen content and shoot nitrogen content, each pondered by the dry biomass of the appropriate compartment. Finally, Nitrogen Uptake Efficiency (NUpE) and Nitrogen Utilization Efficiency (NUtE) were calculated respectively as (i) the ratio of the amount of N absorbed (Plant QN) to the amount of N provided by irrigation (Supplied QN) and (ii) the ratio of the total dry biomass (TDB) to the amount of N absorbed (Plant QN), according to the following equations [44]:

$$NUpE = \frac{Plant\ QN}{Supplied\ QN} \tag{1}$$

$$NUtE = \frac{TDB}{Plant\ QN} \tag{2}$$

The six RSA traits considered in our experiment were previously described in [45] and [20]. The minimal diameter (Dmin) was estimated as the mean of the 15 lowest diameters measured on the roots that visually appeared as the thinnest among the thin root samples. Measurements carried out on the thick root samples led to the estimation of four other RSA traits. The maximal diameter (Dmax) was estimated by measuring 10 diameters along the primary roots and by selecting the maximal value. The inter-branch distance (IBD) was estimated by averaging 50 measures of the distance between two consecutive lateral branches all along the primary root. DlDm is the slope of the linear regression of the diameter of the lateral roots *vs.* that of their mother root, this linear regression being forced to pass through the coordinates (Dmin, Dmin). DlDm was also estimated through 50 measures of lateral root diameter carried out all along the primary roots. VarD is the variation coefficient of the lateral root diameters estimated for 50 measures. The last RSA trait considered in our study is ELT, calculated as the ratio between the elongation rate of the primary root and its maximal diameter. The elongation rate of the primary root was estimated by dividing the length of the primary root by the sum of degree-days at harvest.

For each trait, we considered two biological replicates per modality (GxN). Some traits were measured at the plant scale (LA, TDB, Dmin, Dmax, DlDm, IBD, VarD), hence, values were considered as pseudo-repetitions and were then averaged by tube (n = 32 samples, with values corresponding to the mean of the two plants within the same tube). Because it was impossible to separate the two root systems in the same tube without breaking many roots, other traits (RDB, RS, RTD, CC, NC, NUpE, NUtE and ELT) were directly estimated at the tube scale, *i.e.* without distinguishing the two plants within the same tube (n = 32 samples, with values corresponding to pooled plants).

## 2.5 Data analyses

The mean, standard deviation and coefficient of variation were calculated on each trait, either per N treatment (S2 Table), or per genotype for each N treatment (S3 and S4 Tables). Pairwise correlations were performed on each N treatment for all traits and quantified by using Person's correlation coefficients (S3 Fig, S5 and S6 Tables). Analyses of covariances (ANCOVA) were carried out for each trait to assess the effect of genotype ($G_i$) and nitrogen treatment ($N_j$) on the considered trait. As we were not able to harvest all plants at the same time, the thermal time between sowing and harvest ($TT_lx$) was added as a cofactor in each ANCOVA. Finally, we also considered the interaction between genotype and nitrogen treatments ($GN_{ij}$) and the interaction between the thermal time between sowing and harvest, and the nitrogen treatment ($TTN_{jl}$) in the model (Eq 3).

$$Y_{ijklm} = G_i + N_j + TT_lx + GN_{ij} + TTN_{jl} + \varepsilon_{ijklm} \tag{3}$$

Homocedasticity and residuals normality were verified for each trait and respected for nine traits (Dmin, Dmax, IBD, Dldm, RTD, CC, NC, NUpE and NUtE). Logarithm transformations were used for the six other traits (TDB, LA, VarD, RDB, RS, NUte) to respect homocedasticity and residuals normality conditions. The effect of TT was quantified for each trait using the slope estimated by the ANCOVA (Table 3) considering (i) a linear relation between the trait and TT, (ii) the interaction between TT and the level of nitrogen nutrition, (iii) that TT would have the same impact on each genotype. This quantification was integrated to each trait value, based on the hypothesis that all plants would have been harvested on the same date in the middle of the harvest campaign.

In addition to the significance of G and N effects, the intensity of genotype variations was estimated for each trait by calculating the coefficient of variation among all genotypes. Two Principal Component Analyses (one for PA traits and one for RSA traits) were carried out to synthesize the relationships between traits and evaluate N effect on traits.

In our study, we consider plasticity to N availability as the ability of one genotype to exhibit different trait values in two contrasted N environments [46]. Thus, we calculated two kinds of plasticity for each trait: absolute plasticity (APl$_T$, Eq 4), that was estimated regardless of the variation direction of the plasticity and by pooling all genotypes, and oriented plasticity (OPl$_T$ Eq 5), that considered variation direction of the plasticity and discriminated between genotypes.

$$APl_T = \left| \frac{Y_T^{N+} - Y_T^{N-}}{Y_T^{N+}} \right| \times 100 \tag{4}$$

$$OPl_T = \frac{Y_T^{N+} - Y_T^{N-}}{Y_T^{N+}} \times 100 \tag{5}$$

**Table 3. Covariance analyses (ANCOVA) carried out on each trait to evaluate the effect of genotype (G), nitrogen (N), thermal time (TT), interaction between genotype and nitrogen (GxN) and between nitrogen and thermal time (NxTT).**

| Trait | Type | Transformation | Genotype | Nitrogen | Thermal time | G x N | N x TT |
|-------|------|----------------|----------|----------|--------------|-------|--------|
| TDB | PA | log | 0.01 (**) | 0.82 (***) | 0.17 (***) | 0 (ns) | 0 (ns) |
| RDB | PA | log | 0.01 (ns) | 0.78 (***) | 0.2 (***) | 0 (ns) | 0 (ns) |
| LA | PA | log | 0 (**) | 0.92 (***) | 0.07 (***) | 0 (ns) | 0 (ns) |
| RS | PA | log | 0.02 (ns) | 0.87 (***) | 0.04 (ns) | 0.03 (ns) | 0.02 (ns) |
| RTD | PA | none | 0.03 (ns) | 0.49 (*) | 0.11 (ns) | 0.08 (ns) | 0.22 (ns) |
| NUtE | PA | log | 0.01 (ns) | 0.64 (***) | 0.32 (***) | 0 (ns) | 0.02 (*) |
| NUpE | PA | none | 0.08 (ns) | 0.58 (**) | 0.18 (ns) | 0.09 (ns) | 0.01 (ns) |
| CC | PA | none | 0.11 (ns) | 0.6 (**) | 0.21 (*) | 0.02 (ns) | 0 (ns) |
| NC | PA | none | 0.01 (*) | 0.67 (***) | 0.31 (***) | 0 (ns) | 0 (ns) |
| Dmin | RSA | none | 0.46 (***) | 0.06 (ns) | 0.35 (*) | 0.04 (ns) | 0.03 (ns) |
| Dmax | RSA | none | 0.09 (**) | 0.52 (***) | 0.3 (***) | 0.02 (ns) | 0.06 (*) |
| IBD | RSA | none | 0.05 (**) | 0.75 (***) | 0.14 (**) | 0.02 (ns) | 0.03 (ns) |
| DlDm | RSA | none | 0.16 (**) | 0.54 (***) | 0.22 (**) | 0.04 (ns) | 0.01 (ns) |
| VarD | RSA | log | 0.12 (*) | 0.74 (***) | 0 (ns) | 0.08 (ns) | 0.01 (ns) |
| ELT | RSA | none | 0.23 (ns) | 0.02 (ns) | 0.02 (ns) | 0.15 (ns) | 0 (ns) |

The number in each cell represents a quantification of the impact of the effect (ratio between the mean square of the effect and the total sum of squares). The column "Transformation" indicates which traits were log-transformed before covariance analyses. The number of stars indicates the significance level (***: pvalue < 0.001; **: pvalue < 0.01; *: pvalue < 0.05; ns: non-significant effect).

## 3. Results

### 3.1 Effect of genotype and nitrogen availability on traits

To quantify the effect of N treatments and genotype diversity, ANCOVAs were performed on each of the PA and RSA traits (Table 3). Moreover, the effect of N treatments and genotype diversity were illustrated by boxplots and barplots on eight traits (3 PA traits and 5 RSA traits, Figs 1 and 2; for all 15 traits, see S1 and S2 Figs).

As expected, PA traits were highly sensitive to N nutrition. Multiple ANCOVAs revealed that N availability had a significant effect (pvalue < 0.01) on all the nine plant allocation traits (Table 3). N+ plants were characterized by significantly higher total dry biomass (TDB) and significantly lower root:shoot ratio (RS) and nitrogen uptake efficiency (NUpE) than N- plants (Fig 1). RSA traits were also sensitive to N availability but less than PA traits. N availability had a significant effect on four of the six RSA traits (Table 3). N+ plants were characterized by significantly higher Dmax and DlDm and significantly lower IBD and VarD than N- plants (Fig 1). In contrast, Dmin and ELT showed no significant response to N availability. Using the ratio between the mean square of an effect and the sum of the square of all effects (Table 3), PA and RSA traits were categorised depending on their relative sensitivity either to N availability or to genotype. Thus, (i) LA, RS, TDB, IBD, VarD, DBR and NC were shown to be extremely sensitive to N availability, (ii) CC, NUtE, NUpE, Dldm, Dmax and RTD were more sensitive to N availability than to genotypic variation, (iii) Dmin was more sensitive to genotype variation than to N availability and (iv) ELT was sensitive neither to genotype variation nor to N availability.

The eight genotypes were chosen to maximize the genetic diversity of oilseed rape. However, variations due to genotype were lower than variations due to N treatments on PA traits, as multiple ANCOVAs revealed a significant genotype effect on only three of the nine PA traits. In contrast, significant genotype effect was found on a larger number of RSA traits (five of the six RSA traits (Table 3)) compared to N effects. The amplitude of variations due to

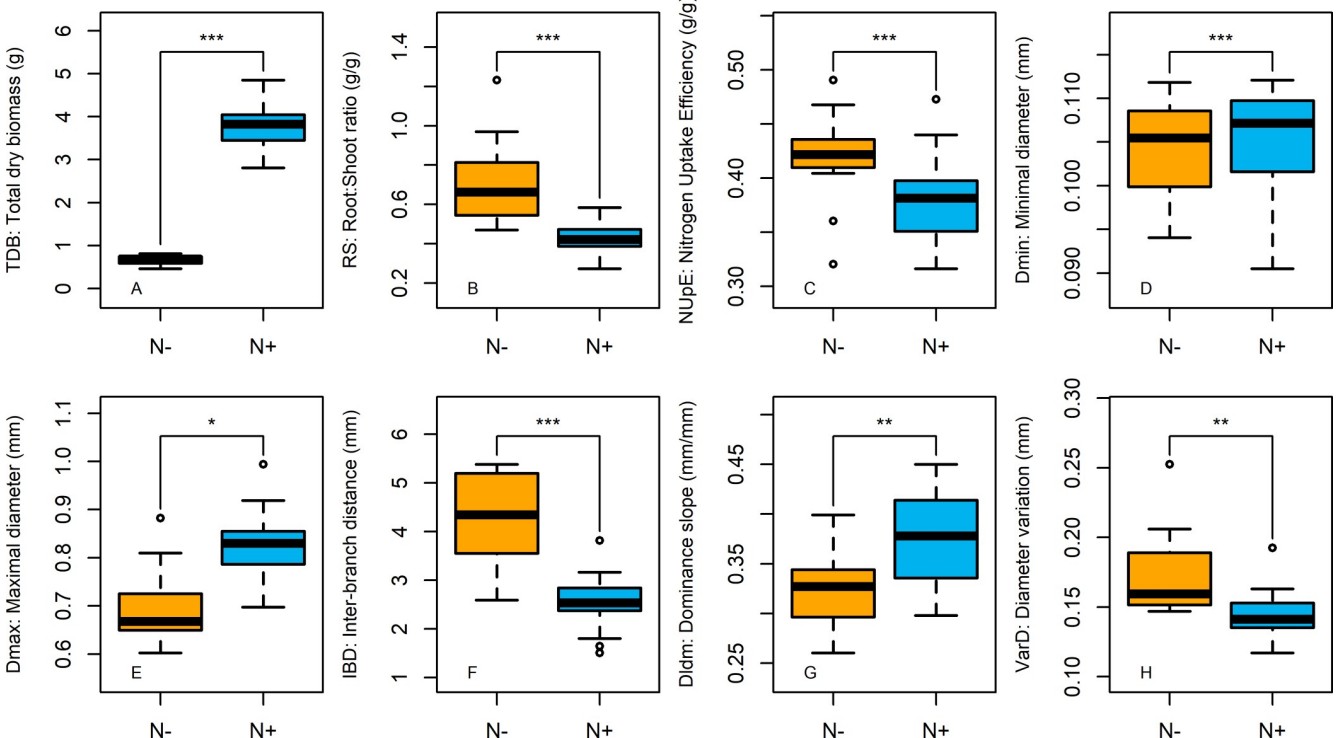

**Fig 1. Effect of N availability on 5 RSA traits and 3 PA traits.** Boxplots showing the effect of the two nitrogen nutritions on total dry biomass (A), Root: Shoot ratio (B), Nitrogen Uptake Efficiency (C), root minimal diameter (D), root maximal diameter (E), Inter-Branch distance(F), root system dominance (G) and lateral root diameter variation (H). Significant differences between the two N treatments were listed on the top of each graph and were determined through an ANCOVA (***: pvalue < 0.001; **: pvalue < 0.01; *: pvalue < 0.05; ns: pvalue > 0.05). Orange boxes stand for N- plants and blue boxes stand for N+ plants (n = 16).

genotype diversity was then compared between PA and RSA traits or between N treatments by calculating the coefficient of variation (cv) of the traits. Overall, genotype variations were more pronounced in N- plants (cv = 26.5 7%) than in N+ plants (cv = 15.15%). In addition, a higher amplitude was found on PA traits compared to RSA traits, under both N treatments (cv = 16% *vs.* 13% for PA and RSA traits, respectively in N+ plants, and 32% *vs.* 18% in N- plants). Finally, no significant G x N interaction was found (Table 3).

### 3.2 Integrated analysis of the response of PA and RSA traits to N availability

PA and RSA traits were studied separately with a Principal Component Analysis (PCA), in order to identify correlations between traits and the overall response of traits to N availability.

The first plane of the PCA performed on PA traits (Fig 3A) was composed of two axes that respectively totalled 68.5% and 15.5% of the variability. The first axis was mainly defined by LA, NC, TDB, RDB and CC on one side and NUtE and RS on the other, while the second axis was mainly defined by NUpE on one side and RTD on the other. The 95% confidence ellipses representing either N+ or N- plants were very well distinguished along the first axis. Thus, N + plants were characterized by high values of TDB, RDB, LA, NC, and low values of NUtE, RS with the opposite trend for N- plants. The second axis was less important for the discrimination of N+ *vs.* N- plants. However, N- plants were characterized by more diverse and higher values of NUpE than N+ plants.

The first plane of the PCA performed on RSA traits (Fig 3B) was composed of two axes that respectively totalled 42.8% and 21.7% of the variability. The first axis was mainly defined by

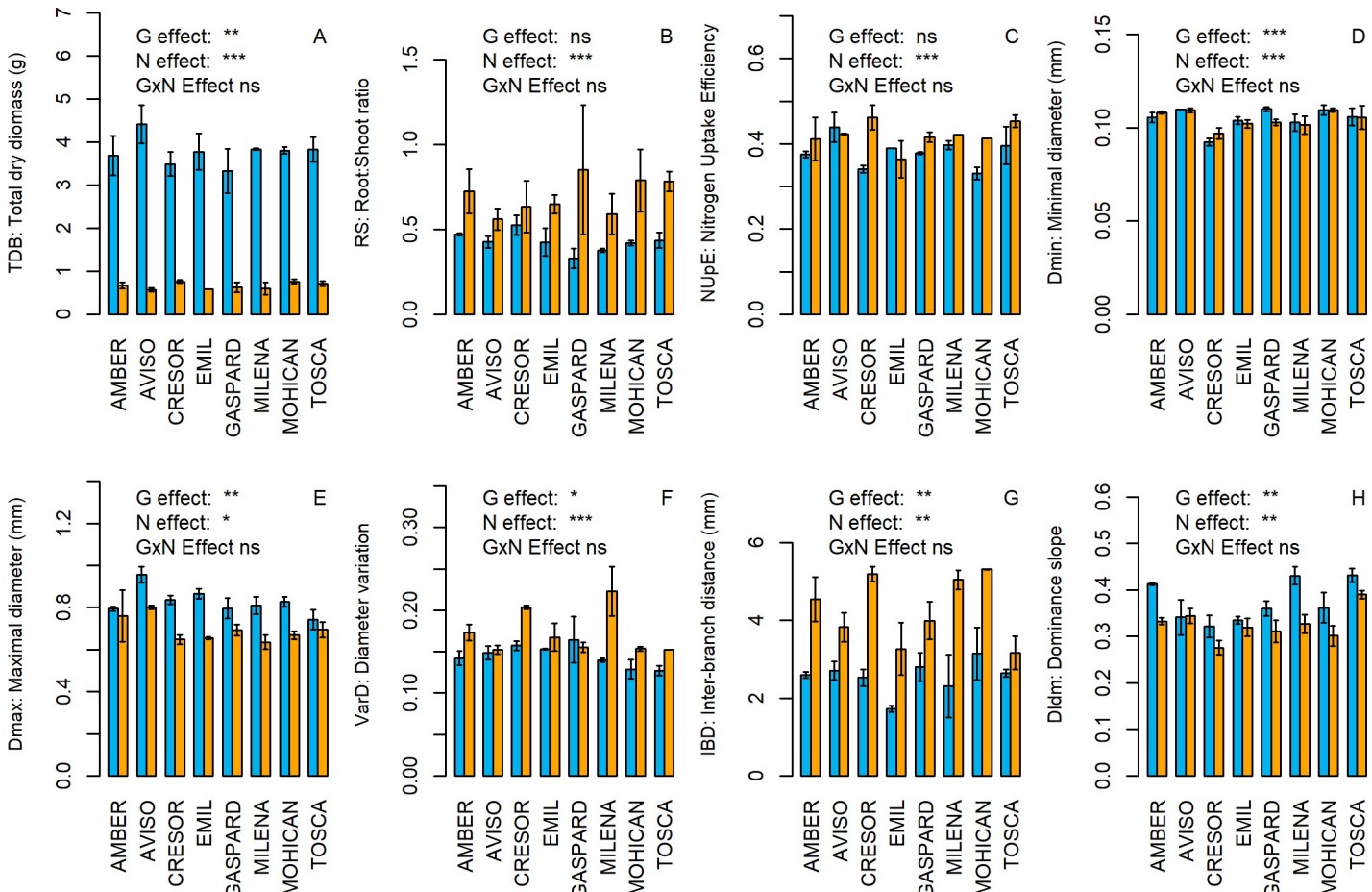

**Fig 2. Genotypic variability of 5 RSA traits and 3 PA traits under N+ and N- treatments.** Barplots showing mean trait values for each genotype under the two nitrogen levels on total dry biomass (A), Root:Shoot ratio (B), Nitrogen Uptake Efficiency (C), root minimal diameter (D), root maximal diameter (E), Inter-Branch distance(F), root system dominance (G) and lateral root diameter variation (H). Significant effect of genotype, nitrogen treatment and interaction between genotype and nitrogen treatment were assessed through an ANCOVA (***: pvalue < 0.001; **: pvalue < 0.01; *: pvalue < 0.05; ns: pvalue > 0.05). Orange bars stand for N- plants and blue bars stand for N+ plants (n = 2). Error bars represent standard errors (+/- SE).

IBD and VarD on one side and Dmax and DlDm on the other, while the second axis was mainly defined by ELT. Dmin was not well represented by any of the two axes. The 95% confidence ellipse representing N+ plants was included in the 95% confidence interval representing N- plants. This indicated a differentiation between N+ and N- plants, that appeared only on the first PCA-component, with N+ plants having higher mean values of Dmax, DlDm and lower mean values of IBD and VarD than N- plants. N+ plants were also characterized by less variability of DlDm, Dmax, VarD and IBD compared to N- plants. The first PCA component also highlighted negative correlations between IBD and Dmax (r = -0.65, pvalue = $2.47 \ 10^{-10}$) and between IBD and DlDm (r = -0.59, pvalue = $9.40 \ 10^{-8}$). Finally, the differentiation between N+ and N- plants was more pronounced on PA traits than on RSA traits.

The correlations highlighted by the PCA were confirmed by pairwise correlations between traits, showing that the correlation pattern was the same in N+ and N- plants (S3 Fig). Indeed, under both N treatments, many PA traits were correlated with each other, whereas few RSA traits were correlated with each other or with PA traits. In N- plants, significant correlations were highlighted between IBD and DlDm (pvalue = 0.015), and between IBD and VarD (pvalue = 0.047) (S5 Table), indicating that plants with many root branching were characterized by

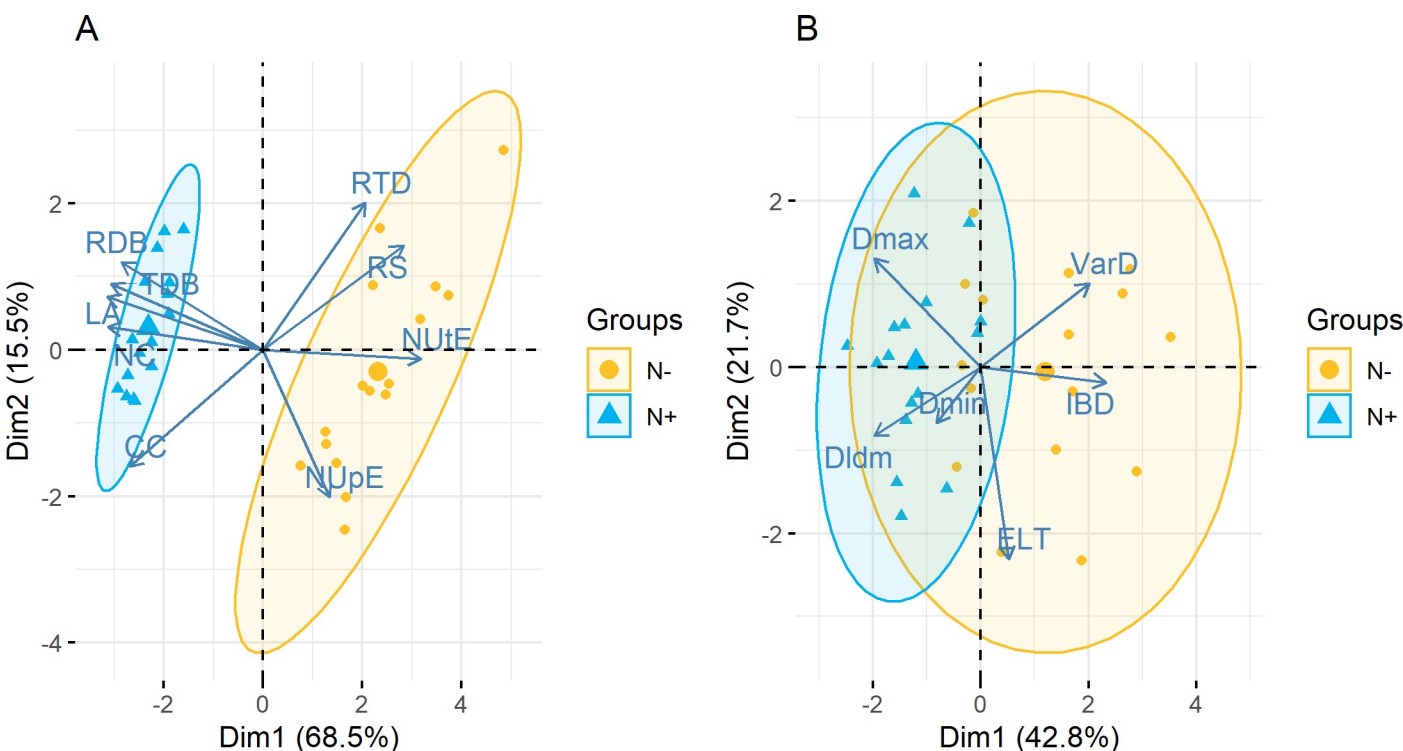

**Fig 3. Principal component analysis.** Trait values and individual positions for PA traits (A) and RSA traits (B) projected on the first plane of the principal component analysis defined by the first two components. RDB: Root Dry Biomass; TDB: Total Dry Biomass; LA: Leaf Area; NC: Nitrogen Content; CC: Carbon Content; RTD: Root Tissue Density; RS: Root:Shoot Ratio; NUtE: Nitrogen Utilization Efficiency; NUpE: Nitrogen Uptake Efficiency; VarD: Lateral root diameter variation; IBD: Inter-Branch distance; ELT: Root elongation rate by root diameter; Dmin: Root minimal diameter; Dldm: Root Hierarchy; Dmax: Root maximal diameter. Colours represent N treatments (blue for N+ plants and orange for N- plants). Biggest dots represent the mean position of the considered group (N+ or N-). Ellipses are two-dimensional projections of the 95% confidence interval.

a low hierarchy between roots of different order and a high variance of diameter variation. In N+ plants, a significant positive correlation (pvalue = 0.023) was highlighted between Dmax and DlDm (S6 Table) indicating that plants with thick roots were characterized by a strong hierarchy between roots of different order.

### 3.3 Trait plasticity in response to N availability

The plasticity of each trait was estimated, first by considering the difference in trait value between N+ and N- plants without discriminating between genotypes, nor taking into account the direction of variation (absolute plasticity), in order to characterize the overall intensity of trait variation in response to N availability. Among the 15 traits studied, six traits were characterized by high absolute plasticity (> 50%). LA and TDB were the most plastic traits (~80%), followed by RDB (~75%), IBD, NUtE and RS (~70%) (Fig 4). NC showed an intermediate absolute plasticity (~40%). Seven traits showed low absolute plasticity (< 20%): RTD, VarD, Dmax, NUpE, DlDm, ELT and CC. Dmin showed nearly no plasticity (near 4%). Except IBD, which was highly plastic, RSA traits generally displayed lowerer absolute plasticity in response to N nutrition than PA traits (~20% and ~51% respectively).

Then, the plasticity of each trait was estimated for each genotype and taking into account the direction of variation (oriented plasticity), which means that positive oriented plasticity stands for higher trait value for N+ plants than for N- plants, and vice versa. Despite the high genotypic diversity previously highlighted on PA and RSA traits, all eight genotypes exhibited the same

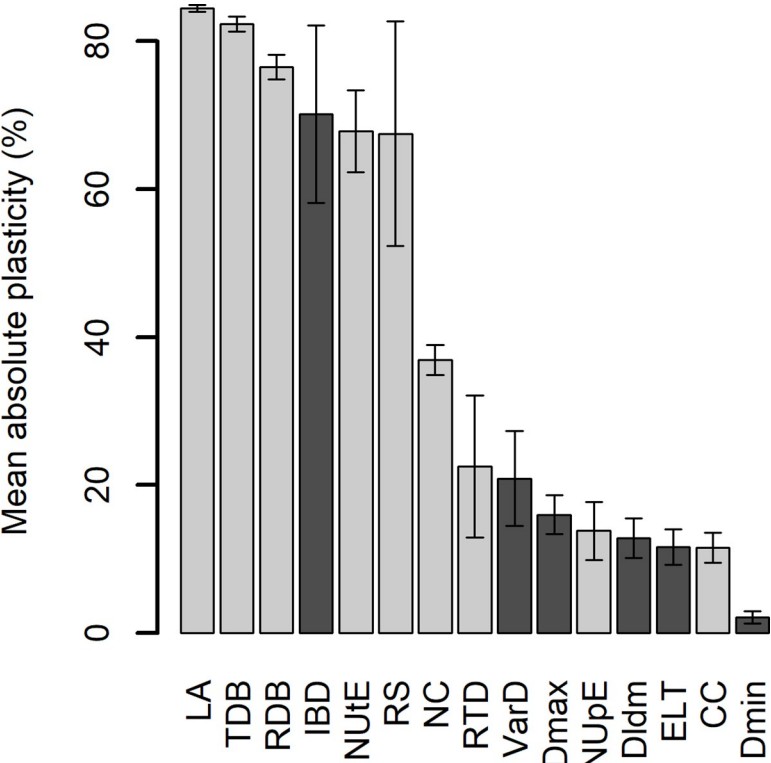

**Fig 4. Absolute plasticity of the 15 measured traits in response to N limitation, averaged by pooling the eight genotypes.** LA: Leaf Area; TDB: Total Dry Biomass; RDB: Root Dry Biomass; IBD: Inter-Branch distance; NUtE: Nitrogen Utilization Efficiency; RS: Root:Shoot Ratio; NC: Nitrogen Content; RTD: Root Tissue Density; VarD: Lateral root diameter variation; Dmax: Root maximal diameter; NUpE: Nitrogen Uptake Efficiency; Dldm: Root Hierarchy; ELT: Root elongation rate by root diameter; CC: Carbon Content; Dmin: Root minimal diameter. Dark bars represent RSA traits while light bars represent PA traits. Error bars are standard errors of the mean (n = 8).

pattern of oriented plasticity, but with various levels of plasticity for each trait ([Fig 5]). Thus, for most genotypes, the plasticity of RS, NUtE and IBD was negative, the plasticity of NC, RDB, TDB and LA was positive, and the plasticity of CC, VarD, ELT, Dmax, DlDm and Dmin was low.

Despite this overall pattern, some genotypes distinguished from others, due to the plasticity of IBD, RTD and RS which differed strongly between genotypes. IBD oriented plasticity was usually strong and negative (between -50% and -115%), but TOSCA exhibited an oriented plasticity for IBD close to 0. TOSCA also distinguished from other genotypes because of very low plasticity level of RSA traits. GASPARD and EMIL were characterized by high RTD plasticity values, while those of MILENA, AVISO and AMBER were very low. GASPARD showed a much higher RS plasticity (~-150%) than other genotypes (~-75%), while that of CRESOR was very low (~-20%). Even if genotypic differences were observed for most traits, no clear strategies were identified between the genotypes.

## 4. Discussion

### 4.1 Using a model framework to study the effect of N availability on various genotypes

In this work, our aim was to quantify the plasticity of the root system architecture to nitrogen availability on a diversified panel of oilseed rape genotypes. To wisely choose the composition of our genotype panel, we had to consider on the one hand the huge amount of experimental

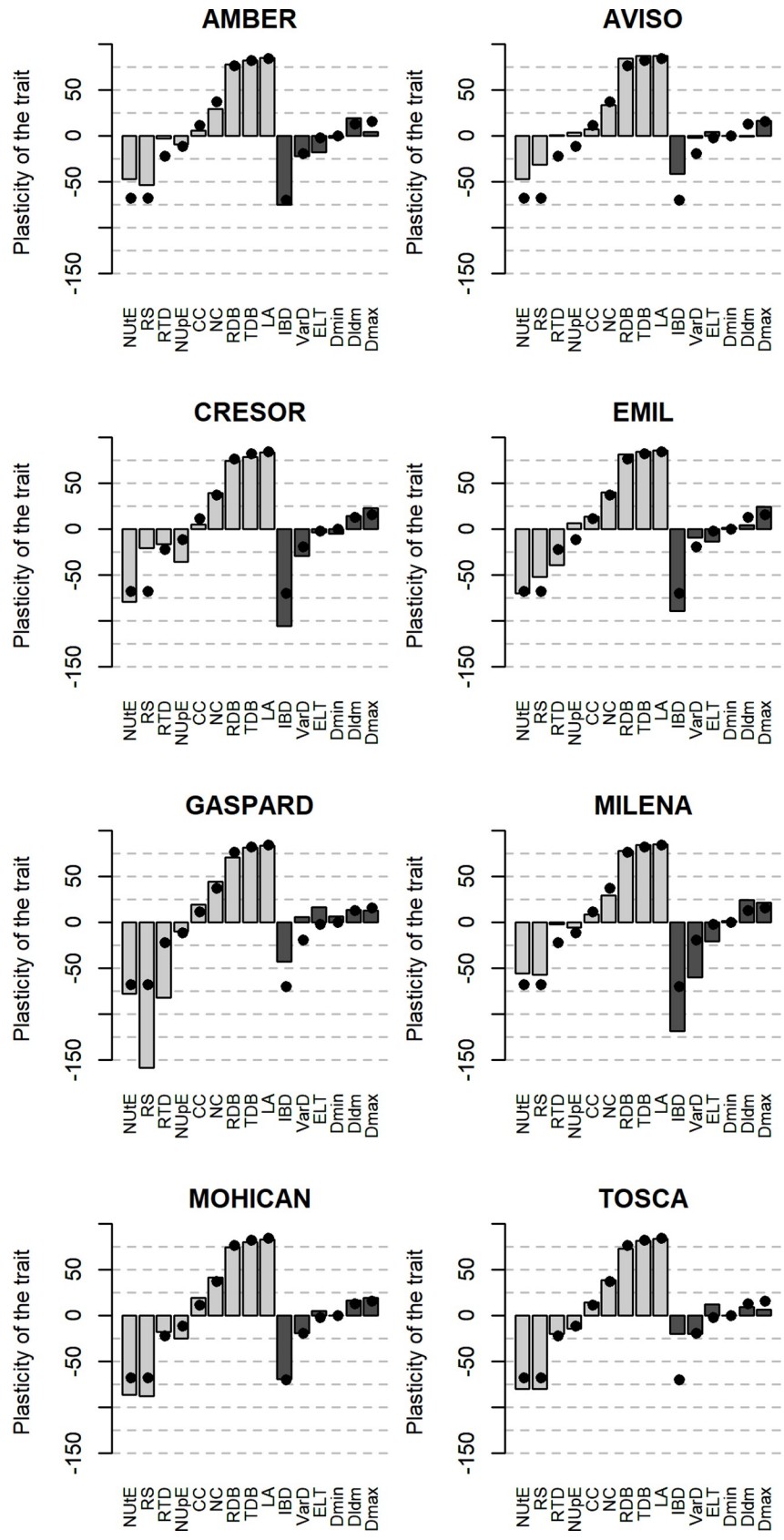

**Fig 5. Oriented plasticity of the 15 measured traits for each genotype.** For each trait, the dark point represents the averaged oriented plasticity calculated by pooling the eight genotypes. NUtE: Nitrogen Utilization Efficiency, RS: Root: Shoot Ratio; RTD: Root Tissue Density; NUpE: Nitrogen Uptake Efficiency; CC: Carbon Content; NC: Nitrogen Content; RDB: Root Dry Biomass; TDB: Total Dry Biomass; LA: Leaf Area; IBD: Inter-Branch distance; VarD: Lateral root diameter variation; ELT: Root elongation rate by root diameter; Dmin: Root minimal diameter; Dldm: Root Hierarchy; Dmax: Root maximal diameter. Dark bars represent RSA traits while light bars represent PA traits.

work (setting up tubes, excavating and spreading each root system, capturing and measuring the scans), and on the other hand our willingness to explore RSA diversity for as many genotypes as possible. To compromise between the two, we chose to limit the number of repetitions to increase the number of studied genotypes. Our bet was successful as more than half of the studied traits were significantly different according to the genotype, and particularly the RSA traits. Trait estimations were accurate enough to also detect strong nitrogen effects using only two replicates, with 13 of the 15 traits impacted by N availability. Total plant biomass was five times higher in non-limited plants compared to N-limited plants, indicating that plants actually experienced contrasting N environments.

At the time of our experiment, phenotypic data were lacking to discrimate between oilseed rape cultivars on RSA. Thus, to maximize the genotypic diversity of RSA within our panel, we hypothesized that the phenotypic diversity of the aerial part could reflect that of the root part. The genotypes were chosen to contrast aerial branching, in addition to seed quality (00/0+ or ++), oilseed rape type (spring/winter), yield performance and stability, geographical origin (ranging from Sweden to France) and finally registration year (ranged from 1981 to 2003), thus accounting for genetic progress. ANCOVAs highlighted that five of the six RSA traits considered were significantly affected by the genotype. This indicates that the selection of genotypes based on aerial and genetic characteristics allowed us to obtain significant genetic diversity in RSA. We hypothesize that this panel represents a significant part of rapeseed diversity, however, this should be confirmed by using panels with more genotypes.

Our approach to characterize RSA was rather innovative. Instead of using only classical integrative traits such as root biomass, root total length, or average root diameter, we chose to use the ArchiSimple model [20], not as a simulation tool, but as a conceptual framework to phenotype RSA, using its parameters as analytical traits. The first main strength of the ArchiSimple parameters is indeed that they are all experimentally measurable. Moreover, measures do not have to be carried out on the whole root system, but only on a few specific aliquots, which allowed us to characterize RSA of plants older than 10-day-old seedling, as it is usually the case [11, 15]. The second main strength is that the ArchiSimple parameters all have a biological meaning. Thus, by phenotyping six of the ArchiSimple parameters, we were able to screen several RSA developmental processes separately. Specifically, we considered the range of root diameters (Dmin and Dmax), the hierarchy between roots (DlDm), the diameter variability of first order ramifications (VarD), the inter-branch distance (IBD) and elongation properties relative to the diameter (ELT). These traits were initially defined to represent the inter-specific variations of RSA among various and contrasting species [45]. To our knowledge, our study is the first to combine an exploration of the intra-specific diversity of RSA with an exploration of the RSA plasticity in response to nitrogen limitation, by phenotyping fifteen traits screening for growth, biomass allocation and model-based RSA.

### 4.2 Genotypic diversity of traits

Even if more than half of the considered traits showed a significant genotypic effect, the effect of N limitation was greater than the genotype effect on plant growth. Indeed, none of PA traits, except NC, TDB or LA, were affected by the genotype. This was probably due to the strong

level of nitrogen constraint we generated. In contrast, all RSA traits (except ELT) were significantly affected by genotype. Dmin appeared to be the most genotypic sensitive RSA trait, as its variation was almost exclusively genetically driven as already suggested by Pagès and Kervella [25]. NUpE reveals the ability of a genotype to absorb nitrogen from the soil and should therefore be directly affected by variations in RSA. However, the absence of a significant genotype effect on NUpE, and of significant correlations between NUpE and RSA traits did not support this hypothesis. Nevertheless, calculations of the power of our statistical tests highlighted that the estimation of genotype effect ought to be underrated due to the small sample size to estimate genotype effect (n = 2). This genotypic variability observed on model-based RSA traits in our experiment is a first step, promising to consider further genetic analyses on a larger panel for breeding purposes.

## 4.3 Plasticity of plant growth

In this study, we characterized plant morphology and biomass allocation through nine PA traits, classically used to characterize plant growth responses to N limitation. Our results are consistent with previous findings indicating a major decrease in plant biomass and leaf area in response to N limitation [47]. This is consistent with the decrease of nitrogen and carbon content in plant tissues under N limitation observed in our experimentation. N-limited plants were also characterized by higher root/shoot ratio, that is consistent with known strategies of biomass allocation in favour of roots under N deficiency [48]. As expected, in our experiment, N- limited plants were more efficient in terms of NUpE and NUtE than N non-limited plants. This is in accordance with previous studies [49], that have formalized that nutrient use efficiency decreases as nutrient availability increases, thus highlighting the potential development of resource economy strategies [50]. However, we highlighted a different response for each of the NUE components. NUtE was indeed more plastic to N availability, consistent with the highest plasticity observed for TDB and LA.

## 4.4. Plasticity of root system architecture traits

The plasticity of RSA is still not well documented and was therefore the central part of our study, considering that RSA traits that would reveal plasticity in response to N availability could be a key toward a better understanding of resource economy strategies. N limitation induced a strong and highly significant decrease in root biomass. It was associated with multiple changes in RSA: root branching density (IBD), variability of root diameters (VarD) and diameter of primary roots (Dmax) were the main traits affected by N limitation.

Some previous works have shown a reduction of average root diameter under nitrogen limitation [51, 52]. However, the model-based RSA traits we used allowed us, for the first time, to go further in the description of the root diameter adaptation in response to N limitation. The diameter of the thinnest roots (Dmin) remained unchanged in response to N limitation. Pagès and Kervella [25] have also observed that the size threshold of the thinnest roots was not sensitive to soil nitrogen content. In contrast, the diameter of the thickest roots (Dmax) decreased in response to N limitation, as well as Dldm (meaning that root hierarchy has increased). Thus, the decrease of average root diameter seems to result from two combined processes: the decrease of the diameter of thicker roots and the tendency for mother roots to produce thinner daughter roots. The overall decrease in root diameters can be seen as an illustration of a more efficient nutrient uptake strategy through an increase in the ratio of root area to root volume. The need for improved efficiency is likely to occur in resource-scarce environments where, for a defined amount of biomass available for the root system, the optimization strategy would be to produce thinner roots to increase exchange area with the soil per root volume. However, as

root elongation rate seems to be linked to apical diameter size [20], a root system composed of thin roots could be penalized by an exploration of a smaller volume of soil and a difficulty to go through compacted soil [53, 54]. Thus, a trade-off might exist between the production of thinner roots to optimize the exchange area with the soil and the production of thicker roots to colonize a greater volume of soil. Nevertheless, previous studies have revealed that the root elongation often increases when nutrients are limited [55], whereas in our study, ELT was not sensitive to N availability. Finally, another major modification of RSA was reflected by the significant increase in IBD (*i.e.* decrease in branching density) due to N limitation. Nitrate ions are known to be mobile in the soil, therefore, a root system with spaced ramification seems to be an optimal way to explore nitrogen-scarce environments. Similar to decreasing root diameters, reducing branching density could be a strategy to save energy allocated to biomass to focus on soil exploration. The strategy would probably not be the same for static nutrients such as phosphorus. Interest in traits related to branching density has increased in recent years [10, 56]. Our study reinforces this interest by showing that branching density was the most sensitive RSA trait to N availability. Therefore, branching density appears as the first lever of RSA adaptation to environmental changes, especially in case of N limitation. This major result highlights that traits related to root branching are critical for nutrient foraging in the soil and therefore of great interest for genetic improvement, especially to develop cultivars adapted to low N input management.

## 5. Conclusion

Using the parameters of a root architecture model as phenotyping traits, our study highlighted the high genetic variability and plasticity of RSA in response to nitrogen availability. Nitrogen limitation mainly resulted in a decrease in the largest root diameters and in lateral root density, associated with a stronger root dominance. Five of the six RSA traits phenotyped showed a significant genotype effect, which is promising for breeding purposes. Among the RSA traits, the inter-branching distance (IBD) is the most promising trait for developing new varieties adapted to low nitrogen inputs, since it showed the highest plasticity in response to nitrogen availability, in the same range as the most plastic PA traits. In the future, the characterization of the ArchiSimple model parameters performed in this study may additionally enable dynamic 3D reconstruction of the root system through model simulations, which may help overcome the bottleneck of high-throughput root system phenotyping.

## Supporting information

**S1 Fig. Pairwise correlations between the 15 measured traits under N+ (up) and N- (bottom) nutritions.** Size and color of the squares indicate the value of the Pearson's correlation coefficient, at significant levels ≤ 0.05. Non-significant correlations are represented by empty cells. PA and RSA traits are shown in black and red, respectively.
(TIF)

**S2 Fig. Effect of the two N nutritions on all 15 measured traits.** Total dry biomass (A), Root dry biomass (B), Leaf area (C), Root:Shoot ratio (D), Root tissue density (E), Nitrogen Utilization Efficiency (F), Nitrogen Uptake Efficiency (G), Carbon content (H), Nitrogen content (I), Root minimal diameter (J), root maximal diameter (K), Inter-Branch distance (L), Root system dominance (M) Lateral root diameter variation (N) and Root elongation rate per root diameter (O). Significant differences between the two N nutritions were listed on the top of each graph and were determined through an ANCOVA (\*\*\*: pvalue < 0.001; \*\*: pvalue < 0.01; \*: pvalue < 0.05; ns: pvalue > 0.05). Orange boxes stand for N- plants and blue boxes stand for N

+ plants (n = 16).
(TIF)

**S3 Fig. Genotypic variability of all 15 traits under N+ and N- treatments.** Barplots showing mean trait values for each genotype under the two N nutritions. Total dry biomass (A), Root dry biomass (B), Leaf area (C), Root:Shoot ratio (D), Root tissue density (E), Nitrogen Utilization Efficiency (F), Nitrogen Uptake Efficiency (G), Carbon content (H), Nitrogen content (I), Root minimal diameter (J), root maximal diameter (K), Inter-Branch distance (L), Root system dominance (M) Lateral root diameter variation (N) and Root elongation rate per root diameter (O). Significant effect of genotype, N treatment and interaction between genotype and N treatment were assessed through an ANCOVA (***: pvalue < 0.001; **: pvalue < 0.01; *: pvalue < 0.05; ns: pvalue > 0.05). Orange bars stand for N- plants and blue bars stand for N + plants (n = 2). Error bars represent standard errors (+/- SE).
(TIF)

**S1 Table. Composition of each of the two nutritive solutions used in the experimentation.**
(DOCX)

**S2 Table. Descriptive statistics of all traits.** Mean, standard deviation (sd) and coefficient of variation (cv) were calculated per N treatment (n = 16).
(DOCX)

**S3 Table. Descriptive statistics of N+ plants.** Mean, standard deviation (sd) and variation coefficient (cv) were calculated per genotype for each trait (n = 2).
(DOCX)

**S4 Table. Descriptive statistics of N- plants.** Mean, standard deviation (sd) and variation coefficient (cv) were calculated per genotype for each trait (n = 2).
(DOCX)

**S5 Table. Correlations analysis between all the 15 measured traits in N+ plants.** Pearson correlation coefficients are indicated in each cell, as well as the level of significance of the two by two correlations (***, Pvalue < 0.001; **, Pvalue < 0.01; *, Pvalue < 0.05; ns, non significant).
(DOCX)

**S6 Table. Correlations analysis between all the 15 measured traits in N- plants.** Pearson correlation coefficients are indicated in each cell, as well as the level of significance of the two by two correlations (***, Pvalue < 0.001; **, Pvalue < 0.01; *, Pvalue < 0.05; ns, non significant).
(DOCX)

## Acknowledgments

The authors thank Anne Laperche (INRA Rennes, France) for her valuable help in the selection of the eight accessions of rapeseed and Mathieu Gruau (INRA Thiverval-Grignon, France) for his contribution to this work. We also thank Elise Alix and Bernard Moulin (INRA Le Rheu, France) and the CRB BraCySol for the packaging and shipping of the seeds. We thank Rebecca James for the English revision of our manuscript that has increased its quality.

## Author Contributions

**Conceptualization:** Loïc Pagès, Céline Richard-Molard.

**Data curation:** Christophe Lecarpentier.

**Formal analysis:** Christophe Lecarpentier.

**Funding acquisition:** Céline Richard-Molard.

**Investigation:** Christophe Lecarpentier.

**Methodology:** Christophe Lecarpentier, Loïc Pagès, Céline Richard-Molard.

**Resources:** Loïc Pagès.

**Supervision:** Loïc Pagès, Céline Richard-Molard.

**Validation:** Céline Richard-Molard.

**Writing – original draft:** Christophe Lecarpentier.

**Writing – review & editing:** Christophe Lecarpentier, Loïc Pagès, Céline Richard-Molard.

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
