## [Decision Letter · Decision Letter 0]

1 Oct 2020

PONE-D-20-23535

Genotypic diversity and plasticity of root system architecture to nitrogen availability in oilseed rape

PLOS ONE

Dear Dr. Lecarpentier,

Thank you for submitting your manuscript to PLOS ONE. After careful consideration, we feel that it has merit but does not fully meet PLOS ONE’s publication criteria as it currently stands. Therefore, we invite you to submit a revised version of the manuscript that addresses the points raised during the review process.

We look forward to receiving your revised manuscript.

Kind regards,

Mehdi Rahimi, Ph.D.

Academic Editor

PLOS ONE

Journal Requirements:

Reviewers' comments:

Reviewer's Responses to Questions

**Comments to the Author**

1. Is the manuscript technically sound, and do the data support the conclusions?

Reviewer #1: Yes

Reviewer #2: Yes

Reviewer #3: Partly

2. Has the statistical analysis been performed appropriately and rigorously? 

Reviewer #1: Yes

Reviewer #2: Yes

Reviewer #3: No

3. Have the authors made all data underlying the findings in their manuscript fully available?

Reviewer #1: Yes

Reviewer #2: Yes

Reviewer #3: No

4. Is the manuscript presented in an intelligible fashion and written in standard English?

Reviewer #1: Yes

Reviewer #2: Yes

Reviewer #3: Yes

5. Review Comments to the Author

Reviewer #1: The aim of this study by Lecarpentier et al was to analyze the genetic diversity and plasticity of system architecture to nitrogen availability in oilseed rape using the ArchiSimple model. This study is thus of much importance and manuscript was generally well written. However, my specific comments are as follows:

1. Abstracts should have no paragraphs to my knowledge.

2. The abstract could be clearly written to explain the importance of this study and how beneficial it will be, as well as explain what the results obtained in the study will be used for in subsequent research or future purposes (If anu)

3. Sentences could be concise. Especially in the introduction section. To me, I think the introduction should be rewritten in a concise manner. As it stands now, there is too much information and seems confusing at a point.

4. The last paragraph of the introduction should be checked if possible.

5. Check manuscript for grammatical (especially subject-verb agreement) and typogragraphical errors. For example lines 13, 33, 45, 48, 50, 53, 54, 56, 58, 60, 61, 108, 179, 210, etc.

6. Line 67 - "They".. Who are the "they"?

7. The Hoagland solution was stated with no reference

8. Line 112 - "800 growing degree-days (around 30 days) after sowing and took place during approximately 150 degree-days". Please crosscheck this sentence

9. At the "trait definition section" some words were bolded. It happens in other parts but most are in this sub chapter. Kindly refer to the guide for authors and make sure the correct formatting is done.

10. Font style of reference is different. I may presume this might have been an oversight, so kindly work on that.

Reviewer #2: paper is well concise but please try to make the results more clear. the discussion ins very long so try to add conclusion as well. the figures are designed very well and show good piece of information related to the paper. overall, the paper is very well written

Reviewer #3: 1. In this study, author discuss about genetic diversity and plasticity of root system architecture in oilseed rape, which is suitable for it. However, number of genotype is very less to characterize root system architecture and plasticity response to nitrogen limitation of oilseed rape.

2. In material and methods; please mention the number of biological replicates which author have studied in control and two levels of availability.

3. Author please mention in manuscript text which statistical test were used for find significant effect of genotype, nitrogen and GxN effects.

4. Author mention in “Experimental design” eight seed per tube were sown and remained two plants plants per tube, please mention at which stage removed others plants from the tube.

5. In Table 1: please explain the accession type what is the meaning of “0+”, “00”, and “++“.

6. In Table1: please indicate the actual yield value or the rank based on the yield value of the accession, So readers can easily understand accession yield performance which author were selected for the study.

7. In Table 1: similarly as yield, please indicate the value of yield stability parameters which author used for stability performance of the accessions.

8. In line 108: Reference is not citing in journal format.

9. In Excavation procedure; line115: author mention that it was impossible to separate the root system of the two plants beyond 10 cm depth. Data collected beyond 10 cm depth, will show sampling error because of inappropriate data collection and will also affect the results. If author collected data of each plant separately, will show better results.

10. Author also calculates “Leaf Area Index” (LAI) based on data scanned by flatbed scanner. LAI is better to understand the dimensionless quantity that characterizes plant canopies.

11. Please do correlation analysis to find out the relationship among and between PA and RSA traits at Nitrogen level (N+ and N-), author not discussed what correlations between traits at two nitrogen level.

12. Please cite reference in manuscript for supporting on-screen root sample measurements.

13. Please write the formula in the manuscript to calculate “Nitrogen Uptake Efficiency” and Nitrogen Utilization Efficiency” and cite the references. I think equations in “Moll et al. 1982” article author can use to re-calculate NUpE and NUtP value.

14. Please prepare descriptive statistics table with mean value, cv value, SE etc.

15. In line 190 to 194: Author calculates absolute and oriented plasticity by equations, please cite the reference in manuscript if any.

16. In line 198: Author only illustrates boxplots and barplots for eight traits why not for all traits.

17. In line223: Author write “nitrogen availability had a significant effect (pvalue < 0.01) on all the nine plant allocation traits”, but in the table 3 there is 13 traits which had a significant nitrogen effect. Please re-check.

18. In line 236: Genotype x nitrogen interaction was significant for three traits only:TDB and LA, but in the Table 3 only one trait (IBD) was significant at pvalue < 0.01.

19. In line 381: Please cite the reference in journal format.

20. Please make sure all references are listed as per journal format, there is formatting error.

6. PLOS authors have the option to publish the peer review history of their article (what does this mean?). If published, this will include your full peer review and any attached files.

Reviewer #1: No

Reviewer #2: **Yes: **Rabail Afzal

Reviewer #3: No

---

## [Author Response · Author response to Decision Letter 0]

31 Mar 2021

Reviewer #1:

The aim of this study by Lecarpentier et al was to analyze the genetic diversity and plasticity of system architecture to nitrogen availability in oilseed rape using the ArchiSimple model. This study is thus of much importance and manuscript was generally well written. However, my specific comments are as follows:

1. Abstracts should have no paragraphs to my knowledge.

In the revised version of the manuscript, line breaks have been removed.

2. The abstract could be clearly written to explain the importance of this study and how beneficial it will be, as well as explain what the results obtained in the study will be used for in subsequent research or future purposes (If any)

The abstract has been rewritten to better highlight the environmental and scientific issues, as well as the impacts of our results for breeding purposes.

3. Sentences could be concise. Especially in the introduction section. To me, I think the introduction should be rewritten in a concise manner. As it stands now, there is too much information and seems confusing at a point.

We have rewritten the introduction section, trying to focus on the main points and to use more simple and concise style. 

4. The last paragraph of the introduction should be checked if possible.

The last paragraph has been rewritten and simplified. 

5. Check manuscript for grammatical (especially subject-verb agreement) and typogragraphical errors. For example lines 13, 33, 45, 48, 50, 53, 54, 56, 58, 60, 61, 108, 179, 210, etc.

Typological and grammatical typos were tracked down throughout the manuscript by careful readings, hoping not to have missed too many.

6. Line 67 - "They".. Who are the "they"?

“They” has been replaced by “Results”.

7. The Hoagland solution was stated with no reference

The reference has been added (L.108) and in the reference list. 

8. Line 112 - "800 growing degree-days (around 30 days) after sowing and took place during approximately 150 degree-days". Please crosscheck this sentence

The sentence has been clarified to more clearly distinguish the beginning of the excavation procedure (800 GDD) from its duration (150 GDD). It can now reads (L 126-127) : 

“The excavation procedure began at 800 growing degree-days after sowing (about 30 days) and lasted about 150 growing degree-days.”

9. At the "trait definition section" some words were bolded. It happens in other parts but most are in this sub chapter. Kindly refer to the guide for authors and make sure the correct formatting is done.

We had used bold letters in this section to make the identification of the trait abbreviations used in the manuscript easier to read. But since this was not carried on in the other sections, and in accordance with the instructions to the authors, we have removed it in the 2.4 section. 

10. Font style of reference is different. I may presume this might have been an oversight, so kindly work on that.

The font style of the references has been homogenized to match the one used in the rest of the manuscript.

Reviewer #2:

paper is well concise but please try to make the results more clear. the discussion ins very long so try to add conclusion as well. the figures are designed very well and show good piece of information related to the paper. overall, the paper is very well written

A special effort has been made to clarify the results, either by improving the English style, by adding additional data as supplementary files, or by changing the order of some sentences to make the organization of the result more coherent. In addition we have added a conclusion to the discussion, which has been shortened a bit. 

 

Reviewer #3:

1. In this study, author discuss about genetic diversity and plasticity of root system architecture in oilseed rape, which is suitable for it. However, number of genotype is very less to characterize root system architecture and plasticity response to nitrogen limitation of oilseed rape.

We are aware that we cannot claim to represent the whole rapeseed diversity through 8 genotypes. Due to the experimental burden required to phenotype all the analytical characters of RSA and allocation that were necessary to meaningfully characterize adaptation to N supply, we were not able to characterize a larger number of genotypes, especially on plants older than seedlings. In the literature, studies characterizing many genotypes with such aged plants and with this level of detail on the root system are scarce. 

However, to compensate for the limited number of genotypes, we paid particular attention to maximizing the diversity of the genotype set, by selecting genotypes that contrasted on several different criteria, capturing genetic progress, winter or spring type, geographical origin, seed quality, aerial architectural diversity, and finally yield performance and stability. The eight genotypes chosen were extracted from a panel of 200 genotypes that was grown in the field under two contrasting nitrogen levels in 20 combinations of location and year (see Corlouer et al, 2019), and phenotyped in previous experiments carried out. Thus, our choice was also based on reliable phenotypic data, in addition to the genotype typology data. 

We have added a sentence in the manuscript to point out the criteria for genotype selection (Lxxx) : 

“Eight genotypes of Brassica napus were selected from a panel of 200 genotypes previously grown in the field under 20 combinations of location and year [39], to maximize (i) the phenotypic diversity of aerial architecture, (ii) the diversity of yield performance and (iii) sensitivity to the environment (ie. ability to produce different yields under various location x year conditions). The set of selected genotypes was also designed to represent the genetic diversity of oilseed rape in terms of registration year (1981-2003), thus accounting for genetic progress; oilseed rape type (winter or spring), which differ by bisannual or annual growth cycle; and finally seed quality according to their glucosinolate and erucic acid content (Table 1).”

2. In material and methods; please mention the number of biological replicates which author have studied in control and two levels of availability.

Each modality (GxN) was replicated twice and two plants were considered in each replicate. For half of the traits (LA, TDB, Dmin, Dmax, Dldm, VarD and IBD), measurements were carried out on each of the two plants and then averaged to finally obtain two replicates per GxN modality, with the value of each replicate corresponding to the mean of the two plants. For other traits (RDB, RS, RTD, CC, NC, NUpE, NUtE and ELT), because it was not possible to fully distinguish the two plants, measurements were carried out at the tube level, by pooling the two plants. Here we also obtained two replicates per modality, but with each replicate value corresponding to the mix of the two plants. This has been specified in the Material and Methods section (L182-188). 

3. Author please mention in manuscript text which statistical test were used for find significant effect of genotype, nitrogen and GxN effects.

Significant effects of genotype, nitrogen and GxN were identified using analyses of covariance (ANCOVA) whose model is detailed in Equation 3. We chose to perform statistical tests based on a linear model rather than simpler separate tests to simultaneously account for the effect of genotype and nitrogen on traits. Moreover, we pointed out that the growth duration (thermal time between sowing and harvest) was different for each plant and thus had a significant effect on many traits. That’s why we chose to perform an ANCOVA that allowed us to also account for the growth duration

4. Author mention in “Experimental design” eight seed per tube were sown and remained two plants plants per tube, please mention at which stage removed others plants from the tube.

The plants were thinned 16 days after sowing, when the plants had about 2 unfolded leaves. We have included these details in the text and referred to the BBCH scale to specify the phenological stage of the plants. It now reads (L106-108) : 

“Seedlings were thinned sixteen days after sowing, when plants had two unfolded leaves, so that only two plants remained in each tube on the 24th of March (phenological stage BBCH 12 [40])”

5. In Table 1: please explain the accession type what is the meaning of “0+”, “00”, and “++“.

For clarity, “accession type” has been replaced by “seed quality” and the table caption has been completed with the following sentence: 

“ Seed quality is a qualitative indicator for erucic acid (C22:1) and glucosinolates (GSL) content, with ++: genotypes rich in C22:1 and GSL, 0+: genotypes poor in C22:1 and rich in GSL, and 00, genotypes poor in C22:1 and in GSL”.

6. In Table 1: please indicate the actual yield value or the rank based on the yield value of the accession, So readers can easily understand accession yield performance which author were selected for the study.

Yield values have been added in Table 1 as requested by the reviewer. These values were extracted from field experiments from the RAPSODYN project [39]. 

7. In Table 1: similarly as yield, please indicate the value of yield stability parameters which author used for stability performance of the accessions.

Similarly, in Table 1 the qualitative parameters of yield stability have been replaced by quantitative values of yield variation, expressed by relative standard deviations of yield means among various environments, based on data from the RAPSODYN field experiments.

8. In line 108: Reference is not citing in journal format.

Sorry for this mistake. It has been corrected. 

9. In Excavation procedure; line115: author mention that it was impossible to separate the root system of the two plants beyond 10 cm depth. Data collected beyond 10 cm depth, will show sampling error because of inappropriate data collection and will also affect the results. If author collected data of each plant separately, will show better results.

As said above, it would not have been possible to collect the root system of each plant separately without breaking many roots. Precisely to avoid sampling errors, we chose not to separate the root systems of the two plants, which would have led to great incertitude in the attribution of broken fine roots to either of the two plants within the same tube. So, for traits for which we were not absolutely certain of the reliability of separating the two plants, we found it more reliable and rigorous to generate data on the pooled plants. 

10. Author also calculates “Leaf Area Index” (LAI) based on data scanned by flatbed scanner. LAI is better to understand the dimensionless quantity that characterizes plant canopies.

We calculate the leaf area, not the Leaf Area Index. LAI was indeed not suitable in our case, since our work was carried out at the individual plant scale and not at the canopy scale. LAI is generally used to normalize the leaf area of plant canopies per unit area of soil and is expressed by dividing leaf area by the soil area occupied by the crop. In our experiment, the tubes were uniformly spaced, so that the soil area occupied by each plant was the same. We can therefore consider the leaf area measurements as near-normalized values, and well suited to comparing various genotypes or N treatments.

11. Please do correlation analysis to find out the relationship among and between PA and RSA traits at Nitrogen level (N+ and N-), author not discussed what correlations between traits at two nitrogen level.

We thank the referee for this suggestion that allowed us to focus on structural correlations apart from the effect of plasticity. A correlation analysis per nitrogen treatment has been added as supplemental data (S5 and S6 Tables, S1 Fig.). The method has been presented in the Material and method section, by adding the following sentence (L191-193) : 

“Pairwise correlations were performed on each N treatment for all traits and quantified by using Person’s correlation coefficients (S3 Fig., S5 Fig. and S6 Fig.)”

Data has been commented on in the result section by adding the following sentences (L305-313): 

“The correlations highlighted by the PCA were confirmed by pairwise correlations between traits, showing that the correlation pattern was the same in N+ and N- plants (S3 Fig.). Indeed, under both N treatments, many PA traits were correlated with each other, whereas few RSA traits were correlated with each other or with PA traits. In N- plants, significant correlations were highlighted between IBD and DlDm (pvalue = 0.015), and between IBD and VarD (pvalue = 0.047) (S5 Table), indicating that plants with many root branching were characterized by a low hierarchy between roots of different order and a high variance of diameter variation. In N+ plants, a significant positive correlation (pvalue = 0.023) was highlighted between Dmax and DlDm (S6 Table) indicating that plants with thick roots were characterized by a strong hierarchy between roots of different order.”

12. Please cite reference in manuscript for supporting on-screen root sample measurements.

We have added a reference that supports on-screen roots measurements by mouse clicking : 

43. Pagès L. Branching patterns of root systems: quantitative analysis of the diversity among dicotyledonous species. Annals of Botany. 2014;114(3):591 8. 

13. Please write the formula in the manuscript to calculate “Nitrogen Uptake Efficiency” and Nitrogen Utilization Efficiency” and cite the references. I think equations in “Moll et al. 1982” article author can use to re-calculate NUpE and NUtP value.

The two equations have been added in the manuscript (Lxxx), as well as the reference to Moll et al. 1982, which indeed corresponds to the same way of calculating NUpE and NUtE as ours. 

14. Please prepare descriptive statistics table with mean value, cv value, SE etc.

Two tables with descriptive statistics have been added as supplementary material. 

- The first one (S2 Table) gathers descriptive statistics of the whole population. Mean, standard deviation and variation coefficient were calculated per nitrogen treatment for each trait.

- The second and the third one (S3 and S4 Table) gather descriptive statistics of N+ and N- plants respectively. Mean, standard deviation and variation coefficient were calculated per genotype for each trait.

A sentence has been added in the material and methods section (L190-191) to refer to these news supplementary tables. 

15. In line 190 to 194: Author calculates absolute and oriented plasticity by equations, please cite the reference in manuscript if any.

Phenotypic plasticity has been defined by Schlichting (1986), as the ability of an individual organism to alter its physiology/morphology in response to changes in environmental conditions. The equations we builded up were derived from this definition and were used to quantify the changes observed between the two levels of nitrogen nutrition. This has been clarified in the manuscript and the reference has been added.

16. In line 198: Author only illustrates boxplots and barplots for eight traits why not for all traits.

To limit the number of graphs and figures, we chose to focus on the main traits (that showed variation to nitrogen and/or genotype (biomass, root:shoot ratio, Nitrogen uptake efficiency and five RSA traits). However, we added the boxplots and barplots shown in Figures 1 and 2 for all 15 measured traits as supplementary data (S3 Fig.).

17. In line223: Author write “nitrogen availability had a significant effect (pvalue < 0.01) on all the nine plant allocation traits”, but in the table 3 there is 13 traits which had a significant nitrogen effect. Please re-check.

Table 3 gathered results concerning plant allocation traits as well as root system architecture traits. To avoid confusions we have added a column specifying the type of each trait. So, Table 3 now clearly indicates that nitrogen availability had a significant effect on all the nine plant allocation traits as well as on four root system architecture traits (13 traits).

18. In line 236: Genotype x nitrogen interaction was significant for three traits only:TDB and LA, but in the Table 3 only one trait (IBD) was significant at pvalue < 0.01.

It was a mistake, thank you for noticing it ! It has been corrected. 

19. In line 381: Please cite the reference in journal format.

Sorry for this mistake. It has been corrected.

20. Please make sure all references are listed as per journal format, there is formatting error.

Sorry for this mistake. All the references have been checked to match the Vancouver journal format.

---

## [Decision Letter · Decision Letter 1]

19 Apr 2021

Genotypic diversity and plasticity of root system architecture to nitrogen availability in oilseed rape

PONE-D-20-23535R1

Dear Dr. Lecarpentier,

We’re pleased to inform you that your manuscript has been judged scientifically suitable for publication and will be formally accepted for publication once it meets all outstanding technical requirements.

Kind regards,

Mehdi Rahimi, Ph.D.

Academic Editor

PLOS ONE

Additional Editor Comments (optional):

Reviewers' comments:

Reviewer's Responses to Questions

**Comments to the Author**

1. If the authors have adequately addressed your comments raised in a previous round of review and you feel that this manuscript is now acceptable for publication, you may indicate that here to bypass the “Comments to the Author” section, enter your conflict of interest statement in the “Confidential to Editor” section, and submit your "Accept" recommendation.

Reviewer #1: All comments have been addressed

Reviewer #3: All comments have been addressed

2. Is the manuscript technically sound, and do the data support the conclusions?

Reviewer #1: Yes

Reviewer #3: Yes

3. Has the statistical analysis been performed appropriately and rigorously? 

Reviewer #1: Yes

Reviewer #3: Yes

4. Have the authors made all data underlying the findings in their manuscript fully available?

Reviewer #1: Yes

Reviewer #3: Yes

5. Is the manuscript presented in an intelligible fashion and written in standard English?

Reviewer #1: Yes

Reviewer #3: Yes

6. Review Comments to the Author

Reviewer #1: Authors have thoroughly revised the manuscript and it looks better now. Manuscript is fit to be accepted for publication.

Reviewer #3: The author have been addressed all the comments in revised manuscript and ready for acceptance. All figure must be summited in required image quality for better visualization.

7. PLOS authors have the option to publish the peer review history of their article (what does this mean?). If published, this will include your full peer review and any attached files.

Reviewer #1: No

Reviewer #3: **Yes: **Javed Akhatar, Punjab Agricultural University, Ludhiana, India

---

## [Editor Report · Acceptance letter]

7 May 2021

PONE-D-20-23535R1 

Genotypic diversity and plasticity of root system architecture to nitrogen availability in oilseed rape 

Dear Dr. Lecarpentier:

I'm pleased to inform you that your manuscript has been deemed suitable for publication in PLOS ONE. Congratulations! Your manuscript is now with our production department. 

Kind regards, 

on behalf of

Dr. Mehdi Rahimi 

Academic Editor

PLOS ONE